# The Influence of Body Composition, Lifestyle, and Dietary Components on Adiponectin and Resistin Levels and AR Index in Obese Individuals

**DOI:** 10.3390/ijms26010393

**Published:** 2025-01-04

**Authors:** Ewelina Polak-Szczybyło, Jacek Tabarkiewicz

**Affiliations:** 1Department of Dietetics, Institute of Health Sciences, Medical College of Rzeszow University, University of Rzeszow, 35-959 Rzeszow, Poland; ewpolak@ur.edu.pl; 2Department of Human Immunology, Institute of Medical Sciences, Medical College of Rzeszow University, University of Rzeszow, 35-959 Rzeszow, Poland; 3Laboratory for Translational Research in Medicine, Centre for Innovative Research in Medical and Natural Sciences, Medical College of Rzeszow University, University of Rzeszow, 35-959 Rzeszow, Poland

**Keywords:** low-grade inflammation, obesity, diet, adipocytes, adiponectin, resistin

## Abstract

Adipose tissue of obese people secretes a number of adipokines, including adiponectin and resistin, which have an antagonistic effect on the human metabolism, influencing the pathogenesis of many diseases based on low-grade inflammation. Body composition analysis using bioelectrical impedance analysis (BIA) was performed in 84 adults with obesity, i.e., body mass index (BMI) greater than or equal to 30 kg/m^2^. Serum was collected to analyze the concentration of adiponectin (ApN) and resistin. The subjects additionally completed a food frequency questionnaire FFQ-6 and a three-day food diary. Adiponectin-resistin index (AR index) was calculated. The results show a positive correlation between resistin levels and BMI and subcutaneous fat content. AR index value was also positively associated with the amount of adipose tissue and body mass. Adiponectin level in the serum of the studied individuals decreased with the content of lean tissue. Adiponectin level also decreased with the amount of carbohydrates, amount of starch, and glycemic load of the diet. Resistin decreased in patients who frequently consumed white pasta and red meat, while AR index was positively associated with the amount of white rice and saturated fatty acids (SFAs) and monounsaturated fatty acids (MUFAs) consumed but negatively associated with the frequent consumption of carbohydrates, including starch. Physical activity was negatively correlated with adiponectin levels and AR index. We concluded that body composition significantly influenced serum resistin and adiponectin concentrations the AR index. Dietary components also had a significant effect.

## 1. Introduction

Adiponectin and resistin are adipokines secreted by adipose tissue. They exhibit opposing effects that may alter the metabolic profile of obese individuals and the development of obesity-related diseases. Adiponectin is a hormone released by adipose tissue that has beneficial effects on insulin sensitivity and inflammation. Low levels of adiponectin are associated with obesity and with several diseases, including type 2 diabetes and atherosclerosis. It is produced primarily by fat tissue cells. Smaller amounts of this adipokine are produced by skeletal muscle cells, cardiac muscle cells, and endothelial cells. Reduced levels of plasma adiponectin are found in obese individuals, especially those with visceral obesity [1]. Resistin is a polypeptide that was described in 2001; unlike adiponectin, it circulates in the body in low concentrations. In humans, the main source of resistin is cells of the immune system: peripheral blood inflammatory cells, monocytes, macrophages, and leukocytes. The level of this adipocine is elevated in obese people, with type 2 diabetes, multiple sclerosis, insulin resistance, and atherosclerosis [1,2]. In a state of hunger, the resistin content is lower than in a state of satiety. In obese people, when obesity is the result of a high-calorie diet and also when it is caused by genetic disorders, its level is higher [3].

The adiponectin–resistin index, which takes into account the levels of both these adipokines, is a promising predictor of metabolic homeostasis and metabolic disorders [1,4]. The index values are lower when the adiponectin level is lower in relation to the high resistin level. In some studies, obesity is associated with an increased AR index [5]. Lau CH et al. suggested considering the AR index as a metabolic risk in obesity and predicting the development of complications related to excess body weight due to the assessment of the opposing effects of both adipokines. According to the authors, the AR index was more strongly associated with an increased risk of type 2 diabetes and metabolic syndrome than ApN and resistin values separately [4]. Some studies confirm that the AR index may be more strongly associated with the prediction of cardiovascular diseases than the level of adiponectin and resistin alone [6,7]. According to the study by Wu O. et al., the AR index was more strongly associated with an increased risk of obesity-related hypertension than adiponectin and resistin separately. This index may be useful in the early diagnosis of obesity-related hypertension [8]. There are many benefits to using the AR index. Among them, it is a cost-effective method that requires only blood collection. Additionally, combined with other routine metabolic assessments, it can detect metabolic syndrome (MS) earlier, help personalize treatment, and increase patients’ chances of recovery or avoiding further complications [4,9].

The levels of both of these adipokines are inextricably linked to the amount of adipose tissue. Obesity is an increasingly widespread problem and results from the complex interaction of various factors in each person (health, genes, diet, metabolism, and physical activity). The increase in adipose tissue mass causes low-grade inflammation, which affects the entire body and is the result of increased secretion of adipokines from preadipocytes and macrophages [10]. Adipose tissue, especially white, is not only an energy store, but also an important regulator of metabolic pathways. Pro-inflammatory and anti-inflammatory factors released by adipose tissue include adipokines such as leptin, adiponectin, resistin, and visfatin, as well as cytokines (IL-1, IL-6, IL-10, IL-17, IL-21, IL-22, IL-23). It is low-grade inflammation that is associated with the consequences of obesity in the form of numerous comorbidities [11,12]. Visceral obesity is considered the main cause. Visceral adipose tissue (VAT) also secretes the anti-inflammatory adiponectin, which has a pleiotropic effect on numerous physiological processes [13]. It is believed that VAT is more strongly associated with a higher risk of metabolic diseases than white adipose tissue. Resistin is not only produced in excessive amounts in obesity, but it itself induces its development and vascular complications, especially atherosclerosis [13].

We hypothesized that lifestyle (eating habits, physical activity, stimulants) significantly affects adiponectin and resistin levels, as well as AR index. Due to the large amount of data that could be obtained in the study, we identified several aims of the study.

Examination of the correlation between BMI levels and adiponectin and resistin and the AR index in obese individuals.Assessment of the effect of individual body composition components on the level of the studied adipokines.Determination of the relationship between the declared comorbidities of the studied subjects and the drugs taken and the level of adiponectin, resistin, and AR index.Examination of the relationship between some dietary habits, addictions, and physical activity with the studied adipokines.Analysis of the correlation of food and food compounds on the level of the studied adipokines and AR index.

The following studies fill the gap in the literature and outline a possible direction for future research. To the best of our knowledge, there are currently no studies investigating dietary and lifestyle factors and their influence on the AR index. This may be important in the appropriate selection of diet therapy, supplementation, or physical activity in improving the above-mentioned indicators.

## 2. Results

Samples from all subjects were analyzed. The values for the entire study group are presented in Table 1, and those grouped into individual obesity classes are presented in Table 2.

Data related to the concentrations of adiponectin and resistin, as well as the results of body composition analysis, were analyzed using the Spearman rank correlation test. Resistin level showed a positive correlation with BMI, total body fat [% and kg], and limb body fat [% and kg]. Negative correlations were found between adiponectin level, AR index, and total and segmental lean body mass—limbs, total, and segmental muscle mass—as well as limbs and trunk, bone mass, and total body water content [kg]. The Spearman’s rank correlation coefficient is listed in the columns, and for statistically significant correlations, it is followed by the *p*-value (Table 3).

The points on the scatterplots (Figure 1), representing subsequent individuals, are arranged in the area of the regression line upwards. This means that an increase in the value of one variable causes a simultaneous increase in the value of the other variable—in this case, with an increase in BMI, the resistin value in the serum of the subjects increases.

Statistical analysis of the relationships between the used medicines and supplements and the serum adiponectin or resistin levels and AR index was performed. Hence, adiponectin levels and AR index values were significantly higher in individuals taking anticoagulants (R = 0.23, *p* = 0.037) and lower in people taking vitamin D_3_ (R = −0.26, *p* = 0.015). Also, adiponectin levels were higher in serum of subjects treated with anticoagulants (R = 2.19, *p* = 0.028) and lower in the serum of patents taking nonsteroidal anti-inflammatory drugs (NSAIDs) on demand (R = −1.98, *p* = 0.048) and supplementing vitamin D3 occasionally (R = −2.15, *p* = 0.032). Due to the small number of subjects taking anticoagulants (n = 4) and NSAIDs (n = 3), as well as the 11 participants who regularly supplemented vitamin D3 among the studied population, these results were not included in the conclusions, and detailed information on them is available in Appendix A.

The questionnaire included questions about lifestyle, like eating habits, addictions, and physical activity. Patients who declared that they practiced sports had lower adiponectin level and AR index values (Table 4). Positive “R” values indicate those with physical activity, eating habits, or addictions that had higher scores, while lower “R” values indicate those with lower scores in that range. Statistically significant differences between those taking and those not taking medications and supplements are shown in red.

The results of the FFQ-6 survey (frequency of consumption of individual food groups) in the analysis with the level of the measured adipokines in the serum of obese people showed that the consumption of red meat and pasta from white flour was associated with higher resistin values. AR index was positively correlated with white rice consumption. The Spearman’s rank correlation coefficient is listed in the columns in Table 5, and for statistically significant correlations, it is followed by the *p*-value.

A three-day food diary summarized using a diet calculator was analyzed in comparison with adiponectin and resistin levels. Adiponectin was lower in patients consuming large amounts of easily digestible carbohydrates (R = −0.22, *p* = 0.042), e.g., starch (R = −0.27, *p* = 0.13). A high glycemic load was associated with lower serum adiponectrin levels (R = −0.21, *p* = 0.49). AR index was also negatively correlated with digestible carbohydrates (R = −0.22, *p* = 0.044), e.g., starch (R = −0.27, *p* = 0.012), but positively correlated with SFA (R = 0.023, *p* = 0.036) and MUFA (R = 0.22, *p* = 0.044) fatty acids. In Appendix A, the rank correlation coefficient is listed in the columns, and for statistically significant correlations, it is followed by the *p* value.

## 3. Discussion

Although adiponectin and resistin have opposite effects, they play an important role in low-grade inflammation, obesity, and morbidities associated with obesity. Their level depends on many factors such as health, diet, and physical activity, but also the content of fat tissue in the body.

Studies on the Spanish population have shown that the resistin level in lean individuals was on average 959 pg/mL. In patients with excess body weight, it was 1266 pg/mL, and obese patients with insulin resistance had an average resistin level of 1331 pg/mL [14]. In our studies, the average resistin level was 27,981 pg/mL, i.e., 27.98 ng/mL. Resistin increases in people with excessive amounts of adipose tissue, which is its main producer. On the other hand, the concentration of adiponectin in serum decreases with the increase in the content of adipose tissue in the body due to its anti-inflammatory nature. The average adiponectin level in our study in people with obesity was approx. 27 µg/mL. It is estimated that in humans, its level may range from 5 to 30 µg/mL depending on health status, lifestyle, and body mass [15]. AR index in the studied population was 0.23 on average, while in the study by Habib et al., among people with an average BMI of 29 kg/m^2^, it was also 0.23 ± 0.13 [16]. After statistical analysis, no correlation was noted between the level of this index and the BMI of the subjects. On the other hand, correlations with body composition component AR index showed similar correlations to adiponectin and body components. In the above-mentioned study, AR index increased with the amount of adipose tissue, unlike in our study [11]. In the present study, adiponectin was inversely correlated with the total and segmental amount of lean tissue, which occurs in proportion to body fat in people with obesity. Obese subjects have an increased amount of lean tissue in order to support the fat tissue and mobilize higher weight of the body. Considering that the increase in adipose tissue is accompanied by an increase in the amount of lean tissue, the correlation shown in the studies seems justified [17,18,19]. However, completely different results for similar studies appear in the literature. Krausse M. et al. raise the issue of a positive relationship between adiponectin and muscle mass, claiming that it has a direct effect on the functional capacity of muscles, partly through insulin sensitization [20]. According to Liu Y. et al., skeletal muscles can also produce adiponectin [21]. Also, studies by Bhuvanya C. et al. and Agostinis Sobrinho C. et al. have shown that there is a positive correlation between the amount of muscle mass and the level of adiponectin in serum [22,23]. The level of resistin also depends on the amount of adipose tissue and thus the BMI value. This is confirmed by numerous studies indicating that its main reservoir is subcutaneous adipose tissue [5,14,24,25].

Obese individuals and patients diagnosed with cardiovascular disease have lower levels of adiponectin. The literature suggests that this is the result of its anti-inflammatory nature [26]. ApN is an antioxidant molecule that inhibits platelet activation by reducing their oxidative stress. According to studies, the antithrombotic effect of adiponectin is associated with the inhibition of platelet aggregation by increasing the activation of endothelial nitric oxide synthesis and blocking the production of hydrogen peroxide [27,28]. The anti-inflammatory properties of adiponectin explain the positive correlation in our own studies between its concentration and antithrombotic drugs. These drugs additionally improve glycemia by reducing inflammatory cytokines, increasing the amount of ApN. Additionally, in our own studies, vitamin D_3_ supplementation was inversely correlated with adiponectin. Completely opposite reports can be found in the literature. Studies usually document a positive relationship between vitamin D_3_ supplementation and ApN [29,30], while some indicate no relationship [28]. AR index was lower in patients taking anticoagulants and lower in those supplementing vitamin D_3_. In a study by Singh P. et al. on 69 male patients with acute coronary syndrome, the AR index level was significantly higher compared to the control group. In addition, it had a better predictive value than adiponectin or resistin concentration by themselves [7].

Respondents who declared frequent exercise had lower adiponectin and AR index levels compared to physically inactive individuals. This relationship has been studied many times. It has been shown that adiponectin is produced in the adipocytes of subcutaneous adipose tissue, but small amounts of it can be produced in muscles [26]. The results of studies are contradictory and may depend on the time devoted to physical activity, its type (in studies, aerobic exercise is usually performed), or other factors occurring during the intervention. Yatagai T. et al. showed that physical exercise on an ergometer for 60 min 5 times a week in twelve lean men not only lowered the level of adiponectin measured immediately after the exercise, but also increased insulin sensitivity. However, the effect of lowering adiponectin did not persist after a week from the end of the exercise [31]. The same relationship appeared in the study by Jurimae J. et al. on ten rowers [32]. There are many more studies suggesting that physical activity in different groups (obese, type 2 diabetics, people with impaired glucose tolerance) increases the level of adiponectin in serum [33,34,35,36]. Data analysis proved a negative relationship between adiponectin level and physical activity, as well as the frequency of carbohydrate consumption, including starch and the glycemic load of the diet. In our own studies, a decrease in adiponectin level was also associated with a high glycemic load of the food eaten. This is due to the anti-inflammatory function of adiponectin, which is antagonistic to foods high in easily digestible carbohydrates and low in fiber, which increase inflammation in the body [37,38]. Consuming large amounts of simple carbohydrates increases the level of pro-inflammatory interleukin 6 and raises inflammatory markers, i.e., C Reactive Protein (CRP) [39]. Studies by Yang W.S. et al. show that high blood glucose levels promote a decrease in the amount of adiponectin in serum [40]. All the above information proves that not only the quality (Glycemic Index) but also the amount of carbohydrates is important in reducing the level of adipokine [41,42]. AR index showed similar relationships to ApN, and a positive correlation was also proven with the frequency of white rice consumption. It has been proven that there is a significant positive relationship between the level of this indicator and the occurrence of type 2 diabetes [16,43].

Dietary components influencing low resistin levels in the serum of the subjects include frequent consumption of red meat and white pasta. A negative correlation with the frequency of red meat consumption has not been described in the literature. The content of saturated fatty acids, which are present in large amounts in red meat, is a factor that increases resistin levels, especially in obesity [44,45]. Red meat is a source of iron. It turns out that resistin was positively correlated with indicators related to the amount of iron in the body in patients with diabetic kidney disease [46]. Also different from our own results are reports in the literature regarding resistin levels and consumption of simple carbohydrates. There is a strong correlation between resistin levels and insulin resistance [24]. A study on pregnant women with gestational diabetes showed that a healthy diet and thus a normal blood glucose level was associated with a lower serum resistin level compared to women with high blood glucose levels [47].

This study, however, has several limitations, though attempts were made to minimize them. Unfortunately, the study group was small, and an additional disadvantage was the lack of a control group. Patients could fill out the questionnaire subjectively, but the instructions for completing it were provided by researchers who were present during the study and were ready to answer any questions. The question about physical activity referred to the subjective assessment of the respondents, as obesity significantly impedes the performance of exercise at the level recommended for people with a normal body weight. However, this study is important because of the analysis of the relationship between diet and the level of AnP or resistin, as well as AR index. Future studies should focus on reducing low-grade inflammation in the bodies of obese individuals through nutrients and combinations in the amount of adipose tissue. Future work should focus on the possibility of designing an effective dietary or pharmacological intervention that will reduce the occurrence of diseases associated with low-grade inflammation.

## 4. Materials and Methods

### 4.1. Ethics

The Institutional Bioethics Committee of the University of Rzeszow (Resolution No. 4 October 2017) approved the project and the Helsinki Declaration was taken into account. Participants signed a voluntary consent to participate and were informed about the possibility of withdrawing from the research at any time.

### 4.2. Subjects

The study included a group of 84 people: 62 women (73.8%) and 22 men (26.2%), with a BMI of 30 kg/m^2^; or more. The subjects were divided into four age groups to indicate the relationships in the individual groups. In summary, 37 people (44.0%) were between 30 and 44 years old, 29 people (34.5%) were between 45 and 60 years old, 12 people (14.3%) were up to 29 years old, and 6 (7.1%) were over 60 years old. The average body weight of the subjects was 103.01 ± 18.23 kg (from 70.6 kg to 156.4 kg). The average BMI of the subjects was 36.65 ± 5.27 kg/m2 (from 30.0 to 58.1 kg/m^2^). A total of 21 people (25.0%) at the time of the study declared that they suffered from hypertension, 3 of the study participants (3.6%) had fatty liver disease, 6 people (7.1%) were treated for hypothyroidism, and 2 people (2.4%) had atherosclerosis. The study participants mainly took blood pressure-lowering drugs (21 people—25.0%), but less frequently, they also took beta blockers (10 people—11.9%), anticoagulants and levothyroxine (4 people each—4.8%), and NSAIDs on an ad hoc basis (3 people—3.6%). Among the study participants, there were 29 people (34.5%) who occasionally took dietary supplements and 11 people (13.1%) who regularly supplemented vitamin D_3_. The study excluded people with pacemakers or prostheses, minors, pregnant women, people with epilepsy, cancer and chronic inflammation (e.g., arthritis or thyroiditis), those taking steroids, antidiabetics, e.g., metformin, anti-inflammatories, statins, and insulin therapy, people unable to maintain an upright body posture, people who were not fasting and had abstained from stimulants, diuretics, and physical exertion for more than 24 h, and people unable to sign an informed consent form.

### 4.3. Blood Sampling and Laboratory Analysis

Blood for testing was collected in the morning on an empty stomach by qualified personnel. It was then centrifuged for 10 min at 4 °C at a relative centrifugal force of 1000× *g* using a SIGMA centrifuge, model 4-16KS (Osterode, Germany). The serum was divided into 200 µL samples and frozen at −86 °C. When all samples were collected, analysis was performed using commercial kits using the multiplex test method based on the magnetic beads Milliplex MAP Kit from EMD Millipore Corporation (Burlington, MA, USA), and the kit used was Human Adipokine Magnetic Bead Panel 1 (Adiponectin, Resistin).

The adiponectin–resistin index (AR) was calculated using the following formula [4]:AR index = 1 + log 10 (R 0) − log 10 (A 0)

The adiponectin–resistin (AR) index was proposed, taking into account both adiponectin and resistin levels, to provide a better indicator of the severity of low-grade inflammation.

### 4.4. Height and Body Composition Measurements and Lifestyle and Nutrition Assessment

The conditions of height and body composition measurements, lifestyle, and nutrition assessment, as well as the eligibility and disqualification criteria for the study, were described in other publications related to this study [9]. The survey is included in the following Appendix A “Lifestyle and Nutrition Assessment”.

### 4.5. Statistical Analysis

Statistical analysis of data was performed using Statistica 13.1. from StatSoft (Krakow, Poland) using nonparametric tests due to the failure to meet the basic assumptions of parametric tests, i.e., compliance of the distributions of the studied variables with the normal distribution, which were verified using the Shapiro–Wilk W test. The statistical description included the median, minimum, maximum, upper and lower quartile, and the mean and standard deviation. The Mann–Whitney U test was used to assess differences in the average level of a numerical feature in two populations. The correlation of two variables that did not meet the criterion of normal distribution was determined using the Spearman rank correlation coefficient. The level of statistical significance was *p* < 0.05.

## 5. Conclusions

Adiponectin values were higher in people taking anticoagulants and lower in people taking NSAIDs ad hoc or supplementing vitamin D_3_. Regular physical activity was also associated with lower adiponectin levels. Also, anticoagulants, D_3_ supplementation, and physical activity affected the AR index value. Simple carbohydrates were associated with lower adiponectin and AR index levels. Similarly, a large amount of lean tissue, which occurs in proportion to the amount of adipose tissue, was negatively associated with these two indicators. This is associated with the occurrence of low-grade inflammation in obesity. Additionally, together with a high content of subcutaneous adipose tissue, the level of resistin increased among the study participants, confirming the relationship between the number of adipocytes and the intensity of inflammation. The last-mentioned relationship was not demonstrated in relation to visceral adipose tissue.

## Figures and Tables

**Figure 1 ijms-26-00393-f001:**
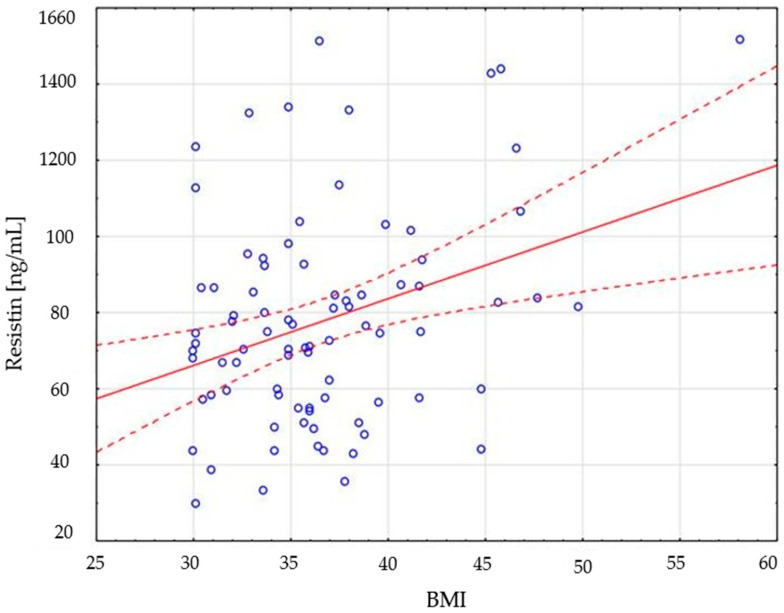
Distribution of resistin values depending on BMI index value.

**Table 1 ijms-26-00393-t001:** Adipokines levels and AR index values.

Adipokines	*n*	$\bar{x}$	SD	Me	Min.	Max.	Q1	Q3
Adiponectin [μg/mL]	84	26.77	44.55	147.16	0.40	317.40	8.72	27.03
Resistin [ng/mL]	84	77.77	27.98	74.65	29.51	151.70	57.32	89.76
AR index	84	0.23	0.64	0.32	−1.58	1.81	0.06	0.56

*n*—number of observations; $\bar{x}$—arithmetic average; SD—standard deviation; Me—median; Min.—minimum; Max.—maximum; Q1—lower quartile; Q3—upper quartile.

**Table 2 ijms-26-00393-t002:** Adipokine levels and AR index values in individual obesity ranges.

	Obesity I Class		Obesity II Class		Obesity III Class	
Adipokines	*n*	$\bar{x}$	*n*	$\bar{x}$	*n*	$\bar{x}$
Adiponectin [μg/mL]	36	23.89	32	36.54	16	13.71
Resistin [ng/mL]	36	74.58	32	72.71	16	95.06
AR index	36	0.099	32	0.288	16	0.049

*n*—number of observations; $\bar{x}$—arithmetic average.

**Table 3 ijms-26-00393-t003:** Correlation between the parameters obtained from the analysis of body composition and adiponectin and resistin levels.

Body Components	Adiponectin [μg/mL]	Resistin [ng/mL]	AR Index
Height [cm]	−0.11	−0.02	−0.12
Body weight [kg]	−0.17	0.18	−0.23 (*p* = 0.036) *
BMI [kg/m^2^]	−0.10	0.22 (*p* = 0.049) *	−0.17
Fat [%]	0.10	0.36 (*p* = 0.001) *	−0.04
Fat [kg]	−0.05	0.28 (*p* = 0.009) *	−0.16
Visceral fat level	0.03	0.07	0.03
Lean body mass [kg]	−0.24 (*p* = 0.028) *	0.03	−0.23 (*p* = 0.037) *
Total water content [kg]	−0.26 (*p* = 0.016) *	0.03	−0.25 (*p* = 0.022) *
Muscle mass [kg]	−0.24 (*p* = 0.027) *	0.03	−0.23 (*p* = 0.036) *
Bone weight [kg]	−0.24 (*p* = 0.025) *	0.03	−0.23 (*p* = 0.033) *
Right leg adipose tissue [%]	0.11	0.29 (*p* = 0.008) *	0.00
Right leg adipose tissue [kg]	−0.02	0.30 (*p* = 0.005) *	−0.15
Right leg lean body mass [kg]	−0.24 (*p* = 0.025) *	0.01	−0.23 (*p* = 0.033) *
Right leg muscle mass [kg]	−0.24 (*p* = 0.026) *	0.01	−0.23 (*p* = 0.033) *
Left leg adipose tissue [%]	0.11	0.26 (*p* = 0.015) *	0.01
Left leg adipose tissue [kg]	−0.02	0.29 (*p* = 0.008) *	−0.14
Left leg lean body mass [kg]	−0.24 (*p* = 0.025) *	0.03	−0.24 (*p* = 0.028) *
Left leg muscle mass [kg]	−0.24 (*p* = 0.027) *	0.03	−0.23 (*p* = 0.032) *
Right hand adipose tissue [%]	0.02	0.28 (*p* = 0.009) *	−0.10
Right hand adipose tissue [kg]	−0.07	0.30 (*p* = 0.006) *	−0.19
Right hand lean body mass [kg]	−0.24 (*p* = 0.028) *	−0.03	−0.20
Right hand muscle mass [kg]	−0.23 (*p* = 0.032) *	−0.04	−0.19
Left hand adipose tissue [%]	0.01	0.28 (*p* = 0.009) *	−0.11
Left hand adipose tissue [kg]	−0.09	0.28 (*p* = 0.010) *	−0.20
Left hand lean body mass [kg]	−0.22 (*p* = 0.040) *	−0.01	−0.20
Left hand muscle mass [kg]	−0.22 (*p* = 0.042) *	−0.01	−0.20
Torso adipose tissue [%]	0.10	0.17	0.03
Torso adipose tissue [kg]	−0.00	0.18	−0.07
Torso lean body mass [kg]	−0.21	0.07	−0.21
Torso muscle mass [kg]	−0.22 (*p* = 0.047) *	0.07	−0.22 (*p* = 0.048) *

* red color indicate significant values (*p* < 0.05).

**Table 4 ijms-26-00393-t004:** Relationship between life style factors and adipokine levels and AR index.

	Adiponectin [μg/mL]		Resistin [ng/mL]		AR Index	
	R	*p*	R	*p*	R	*p*
Tea consumption	0.07	0.518	0.20	0.072	−0.02	0.858
Coffee consumption	0.07	0.513	0.04	0.723	0.06	0.568
Sweetening beverages	−0.08	0.484	−0.12	0.279	−0.06	0.578
Drinking alcohol	−0.06	0.604	0.04	0.694	−0.03	0.775
Smoking cigarettes	−0.05	0.653	−0.21	0.058	0.03	0.786
Over-salting food	0.03	0.771	−0.00	0.993	0.01	0.912
Snacking between meals	−0.09	0.418	0.17	0.113	−0.13	0.221
Physical activity	−0.26	0.015 *	0.08	0.497	−0.26	0.017 *

R—rank correlation coefficient as a result of Spearman test; *p*-level of significance; * red color indicate significant values (*p* < 0.05).

**Table 5 ijms-26-00393-t005:** Correlation of the relationship between the frequency of consumption of selected products and the level of adipokines and AR index.

Adiponectin [μg/mL]		Resistin [ng/mL]	AR index
White bread	−0.05	0.01	0.09
Wholemeal bread	0.09	0.10	−0.05
Confectionery bread	0.05	−0.06	−0.08
Oat flakes, barley, rye	0.07	0.00	−0.03
Buckwheat, barley, millet	−0.11	−0.02	0.13
White rice	−0.18	0.13	0.23 (*p* = 0.036) *
Brown rice	0.17	0.12	−0.13
White noodles	−0.05	−0.22 (*p* = 0.047) *	0.15
Whole-grain pasta	0.13	0.10	−0.13
Potatoes	−0.08	0.05	0.11
Poultry	0.03	−0.10	0.01
Red meat	−0.06	−0.24 (*p* = 0.027) *	0.11
Fish	−0.06	0.16	0.05
Cold cuts, frankfurters, sausages	0.09	−0.01	−0.09
Milk	−0.02	−0.08	0.01
Natural sour milk products	−0.11	0.13	0.15
Sour fruit milk products	0.02	0.21	0.04
Cottage cheese	0.14	0.08	−0.13
Cheese	0.11	0.08	−0.11
Eggs	0.03	0.12	0.02
Vegetables	−0.01	0.05	0.02
Fruit	0.00	0.03	0.02
Cakes, cookies	0.14	−0.01	−0.13
Bars, chocolates, chocolates	0.07	−0.13	−0.12
Juices, nectars, fruit drinks	−0.19	−0.04	0.19
Sweetened carbonated drinks	−0.13	−0.06	0.14
Fast-food dishes	−0.08	0.06	0.09

* red color indicates significant values (*p* < 0.05).

## Data Availability

The data presented in this study are available on request from the corresponding author. The data are not publicly available, as they include sensitive clinical data.

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
