# Peer review of "The Influence of Body Composition, Lifestyle, and Dietary Components on Adiponectin and Resistin Levels and AR Index in Obese Individuals"

_ijms, 2025, doi:10.3390/ijms26010393_

Round 1
Reviewer 1 Report
Comments and Suggestions for Authors
Abstract
Background: the background should place emphasis on the role of diet on adiponectin and resistin as inflammatory markers in metabolic health and briefly mention the importance of studying such associations. There were no results presented how the immune system will be regulated so this portion is quite misleading.
Methods: Authors did not mention how the associations between the risk factors and outcomes were examined.
Results: The title mentions “nutrients” but the results focus largely on dietary components. Might want to consider changing the title.
Results are also hard to read the way it is written. Authors want to consider focusing on the risk factors of body composition, diet and lifestyle on the exposure outcome of adiponectin, resistin and AR index and consolidating some of the findings. For example: “Both BMI and adipose tissue content was positively associated with resistin and AR index value while low lean mass was associated with higher adiponectin levels. (how was this analyzed? And what are the outcomes of the analysis? Was there a pvalue or beta coefficients to report?).
Authors might also want to consider using consistent terms throughout the manuscript such as: fat mass and lean mass to help improve the readability of the manuscript. It is confusing to have many different terms such as BMI, body mass, subcutaneous tissue content, adipose tissue and lean tissue.
Was wondering if is it necessary to mention the role of BMI in these associations when all the subjects examined are overweight and above the BMI of 30.
The results are a bit confusing when AR index was positively associated with white rice but negatively with the consumption of carbohydrates and starches. But isn’t white rice a part of that food group?
Conclusion: Use serum resistin and adiponectin. The conclusion with dietary observations can be more nuanced and meaningful. Were the findings in line with the hypothesis?
Introduction: In the introduction it says that the AR index is a better predictor of metabolic outcomes. In this case, do the authors want to consider just presenting the AR Index and perhaps it would be easier to describe the findings?
Line 71-75: There is no mention here of exploring the role of body composition with adiponectin, resistin and AR index. Also suggest to change dietary factor to dietary components and again to consider changing the title to remove the word nutrients
Results: The Table 1 and 2 adipokines results seem too large to be meaningful – consider presenting in a different unit?
Table 3: Some values are presented with “.” And some with “,”. If the findings with drugs and supplements are not the main findings of the paper I would suggest for this portion to be moved to the supplementary tables. The table also doesn’t seem to be labelled or drawn properly.
Footnote should state that R is an abbreviation.
Table 4: Again, there is inconsistency in the terms used throughout, dietary factors, dietary habits or lifestyle factors. From the introduction there was no indication that dietary habits would be explored as well on top of dietary factors. And how is physical activity a “dietary habit”?? Shouldn’t this table be the lifestyle factors?
Table 5: There is no label for the values presented and again some values were presented with “.” And some “,” . I do not think there is a need to present all the p values but just to use an asterisks to indicate the significant findings.
Figure 2: What is “BMI level”?
And why is it that this paper did not present a characteristics table of the study participants in this study?
Line 244: Materials and methods
I think it very odd and unusual format for the materials and methods to be placed after the discussion of the manuscript.
Line 284: It is fine to refer to other details from another manuscript but perhaps the methods of assessment should still be briefly mentioned in this section.
Discussion
Line 141: The authors should try to summarize the main findings from the many associations that were performed in this study. With all the variables examined it was hard to appreciate the main findings of this study.
Line 150: It is a bit hard to follow what is means here when adiponectin decreases with the increase of adipose tissue and in Line 161 adiponectin also decreases with the increase amount of lean tissue? From what I understand in the introduction, the adiponectin levels should decrease in subjects with higher adipose tissue and thus should be lower in individuals with higher lean mass?
Major comments: The authors should consider tidying up the aims of study to being with and to consider presenting very clear exposure outcomes (eg: adipose tissue, lifestyle factors and dietary components) with A/R index.
Right now there are too many exposure variables and outcomes variables that it makes reading the manuscript very difficult and it is hard to appreciate the findings. The authors need to decide what is it they want to study, the correction with fat mass or lean mass?
The manuscript is also structed strangely with the methods and materials appearing after the discussion portion of the manuscript
The tables were not well drawn, or labelled.
Comments on the Quality of English LanguageAcceptable but the writing of the manuscript needs significant improvement
Author Response
Dear Sir or Madame
We would like to thank the Reviewer for comprehensive and helpful evaluation of our manuscript. The Reviewers concerns about our paper were very helpful and we have revised the paper according to their suggestion. We have implemented most comments of the Reviewers and hereby send you the new version of manuscript as well as a letter including all answers and text changes. We also provide revised manuscript with Track Changes' file.
Your sincerely,
Ewelina Polak-Szczybyło
Jacek Tabarkiewicz
Response to Reviewer 1 Comments
Background:
Point 1. the background should place emphasis on the role of diet on adiponectin and resistin as inflammatory markers in metabolic health and briefly mention the importance of studying such associations. There were no results presented how the immune system will be regulated so this portion is quite misleading.
Response 1: We would like to thank Reviewer for this comment. We have changed this part of the abstract to primarily refer to obesity, adiponectin, and resistin.
Point 2. Methods: Authors did not mention how the associations between the risk factors and outcomes were examined.
Response 2: We only mentioned in the Abstract that the results had been statistically significant, but the “Abstract” must be limited to 200 words. The current one is 243 words. I hope that readers will look into the manuscript for information missing in the “Abstract”.
Point 3. Results:
- a) The title mentions “nutrients” but the results focus largely on dietary components. Might want to consider changing the title.
- b) Results are also hard to read the way it is written. Authors want to consider focusing on the risk factors of body composition, diet and lifestyle on the exposure outcome of adiponectin, resistin and AR index and consolidating some of the findings. For example: “Both BMI and adipose tissue content was positively associated with resistin and AR index value while low lean mass was associated with higher adiponectin levels. (how was this analyzed? And what are the outcomes of the analysis? Was there a p-value or beta coefficients to report?).
- c) Authors might also want to consider using consistent terms throughout the manuscript such as: fat mass and lean mass to help improve the readability of the manuscript. It is confusing to have many different terms such as BMI, body mass, subcutaneous tissue content, adipose tissue and lean tissue.
- d) Was wondering if is it necessary to mention the role of BMI in these associations when all the subjects examined are overweight and above the BMI of 30.
- e) The results are a bit confusing when AR index was positively associated with white rice but negatively with the consumption of carbohydrates and starches. But isn’t white rice a part of that food group?
Response 3:
- a) By definition, dietary component is a broader concept. Replacing nutrients with this term is a good indication.
- b) We did not find the sentence quoted by the reviewer to refer to it directly in manuscript. However, below I will quote the paragraph in “Results” related to the outcomes regarding the analysis of body composition and the values ​​of adiponectin, resistin and AR index. The method of conducting the analysis is mentioned, what are the results of the analysis, and in Table 6 quoted at the end of the paragraph are all the necessary p and R values.
“Data related to the levels of adiponectin and resistin as well as the results of body composition analysis were analyzed using the Spearman rank correlation test. Resistin level showed a positive correlation with BMI, total body fat [% and kg] and limb body fat [% and kg]. Negative correlations were found between adiponectin level, AR index and total and segmental lean body mass - limbs, total and segmental muscle mass - limbs and trunk, bone mass, total body water content [kg]. The Spearman’s rank correlation coefficient is listed in the columns, and for statistically significant correlations, it is followed by the p-value (Table 6).”
- c) Thank you for this comment, but these are completely different terms, not synonyms. Each of them means a completely different component of body mass or anthropometric indicator.
- d) BMI is an anthropometric indicator that takes into account the proportion between body weight and height. According to WHO, it is BMI that determines the degree of obesity, emphasizing the differences between a BMI of 30 kg/m2 and 40 kg/m2. I will also mention that in the study group, the BMI range was between 30 kg/m2 and 58.1 kg/m2 (subsection 4.2). The body components change with this parameter as well as metabolism, amount of the fat cells secreting mediators influencing different biological processes including inflammation.
- f) We change sentence in Abstract about carbohydrates, rice and starch. Products other than rice also contain carbohydrates and starch. The respondents obtained carbohydrates not only from this source. In addition, carbohydrates are determined not only by their quantity but also by their quality, expressed, for example, by the glycemic index.
Point 4. Conclusion: Use serum resistin and adiponectin. The conclusion with dietary observations can be more nuanced and meaningful. Were the findings in line with the hypothesis?
Response 4: The hypothesis was not mentioned in the article and we are agree it had to be corrected. We added the hypothesis before the aims of the study. The results are consistent with the hypothesis.
Introduction:
Point 5. In the introduction it says that the AR index is a better predictor of metabolic outcomes. In this case, do the authors want to consider just presenting the AR Index and perhaps it would be easier to describe the findings?
Response 5. Indeed, AR index is more important in predicting metabolic syndrome than adiponectin and resistin separately. However, our aim was the influence of food components, lifestyle factors and body composition. To present the results in correlation to AR index, we used the results of adiponectin and resistin levels. On the other hand, it can be noticed that significant correlation of the concentrations of resistin and adiponectin are unanimous with correlations of AR index. We have to remember that AR index is mathematical calculations and can expose some strong associations, probably the most important ones. In out paper we decided to show also analysis of raw concentration to have deeper insight.
Point 6. Line 71-75: There is no mention here of exploring the role of body composition with adiponectin, resistin and AR index. Also suggest to change dietary factor to dietary components and again to consider changing the title to remove the word nutrients
Response 6: Line 68-83 are dedicated to information about subcutaneous and visceral adipose tissue in relation to adiponectin and resistin, as well as obesity (BMI status above 30 kg/m2) and these parameters. Line 50-56 are a description of the relationship between obesity and AR index. Due to the small number of studies on AR index, we did not find any reference to the amount of adipose and lean tissue.
We changed the nutrients to dietary components.
Results:
Point 7. The Table 1 and 2 adipokines results seem too large to be meaningful – consider presenting in a different unit?
Response 7: Thank you for that comment. We present them in unit which was used for calculating AR index to be consistent with approved calculation of AR index and other papers considering this parameter.
Point 8. Table 3: Some values are presented with “.” And some with “,”. If the findings with drugs and supplements are not the main findings of the paper I would suggest for this portion to be moved to the supplementary tables. The table also doesn’t seem to be labelled or drawn properly. Footnote should state that R is an abbreviation.
Response 8: This is a very valid comment. We correct that.
Point 9 Table 4: Again, there is inconsistency in the terms used throughout, dietary factors, dietary habits or lifestyle factors. From the introduction there was no indication that dietary habits would be explored as well on top of dietary factors. And how is physical activity a “dietary habit”?? Shouldn’t this table be the lifestyle factors?
Response 9: This is a valid comment. We also correct that.
Point 10 Table 5: There is no label for the values presented and again some values were presented with “.” And some “,” . I do not think there is a need to present all the p values but just to use an asterisks to indicate the significant findings.
Response 9: Thank you for your comment. We have made changes.
Point 11 Figure 2: What is “BMI level”?
Response 11: Thank you for this note. We change it to “BMI index value”.
Point 12 And why is it that this paper did not present a characteristics table of the study participants in this study?
Response 12: Data related to the value of body weight, height, BMI, age, and population size for both sexes are presented in subchapter 4.2. Additionally, information on medications taken and declared diseases can be found there.
Materials and methods:
Point 13 Line 244: Materials and methods I think it very odd and unusual format for the materials and methods to be placed after the discussion of the manuscript.
Response 13: This is due to the template that is mandatory for manuscripts submitted to this journal. It dictates this order of chapters.
Point 14 Line 284: It is fine to refer to other details from another manuscript but perhaps the methods of assessment should still be briefly mentioned in this section.
Response 14: Data for this manuscript come from a larger study on obese individuals. In this project, the measurement conditions and the method of their execution were the same. We decided not to duplicate the descriptions because it is described in sufficient detail in the cited publication.
Discussion:
Point 15. Line 141: The authors should try to summarize the main findings from the many associations that were performed in this study. With all the variables examined it was hard to appreciate the main findings of this study.
Response 15: We have included the main findings in the “Conclusions” section.
Point 16. Line 150: It is a bit hard to follow what is means here when adiponectin decreases with the increase of adipose tissue and in Line 161 adiponectin also decreases with the increase amount of lean tissue? From what I understand in the introduction, the adiponectin levels should decrease in subjects with higher adipose tissue and thus should be lower in individuals with higher lean mass?
Response 16. Thank you for this comment. People with obesity have an increased amount of lean tissue in order to support the fat tissue. Therefore, the relationship between adiponectin and lean tissue may be a result of increasing adipose tissue. This is a possible explanation. We have added another sentence explaining these proportions.
Point 17. Major comments: The authors should consider tidying up the aims of study to being with and to consider presenting very clear exposure outcomes (eg: adipose tissue, lifestyle factors and dietary components) with A/R index.
Response 17: Thank you for your opinion. We have tried to systematize the knowledge as the Reviewer advised us. At the end of the "Introduction" chapter, we have established the aims of the study in the order in which we then present the results and discuss them in the discussion.
Point 18. Right now there are too many exposure variables and outcomes variables that it makes reading the manuscript very difficult and it is hard to appreciate the findings. The authors need to decide what is it they want to study, the correction with fat mass or lean mass?
Response 18: Lean mass and fat mass are two completely different tissues that often have opposing effects, just as adiponectin and resistin are adipokines with opposing effects. It is worth presenting both of these components.
Point 19. The manuscript is also structed strangely with the methods and materials appearing after the discussion portion of the manuscript
Response 19. As we mentioned in "Response 13", this arrangement is dictated by the journal and the manuscript template containing this construction. Other publications in this journal also contain this construction.
Point 20. The tables were not well drawn, or labelled.
Response 20. We tried to unify the tables, add all necessary abbreviations and remove "," in numeric values ​​and insert "."
Reviewer 2 Report
Comments and Suggestions for Authors
The manuscript is acceptable for the scope of the current Special Issue. This research provides interesting correlations between lifestyle factors, physical activity and body composition in obese individuals and the level of adiponectin, resistin and A/R index.
Despite the limitations, which are mentioned in the discussion, the manuscript is well constructed. However, minor improvements should be included, before consideration for publication.
1)Provide additional rationale and evidences in both introduction and discussion why the A/R index is more informative than the levels of adiponectin and resistin alone.
2)There are typos and other mistakes within the text, which should be corrected after careful proofreading of the entire manuscript. Some of them are:
a. You have only one figure, labeled Figure 2, please make a correction or provide rationale for this.
b. Line 60 – one comma missing after [7].
c. Line 69 – what is that X in brackets for references?
d. Row 116 and 119 - p=0042 and p=0044.
e. Please use consistently decimal separator - full stop “.”, not comma “,” doublecheck all tables and the text.
f. Please, introduce all abbreviations when used for first time.
g. On line 120, Supplementary table should be 1, not 2.
h. Format the superscripts within the text properly (e.g. m2).
i. In subsection 4.4 Height and Body Composition Measurements and Lifestyle and Nutrition Assessment, you should refer to Supplementary Materials: File S1 “Lifestyle and Nutrition Assessment”
Author Response
Dear Sir or Madame
We would like to thank the Reviewer for comprehensive and helpful evaluation of our manuscript. The Reviewers concerns about our paper were very helpful and we have revised the paper according to their suggestion. We have implemented most comments of the Reviewers and hereby send you the new version of manuscript as well as a letter including all answers and text changes. We also provide revised manuscript with Track Changes' file.
Your sincerely,
Ewelina Polak-Szczybyło
Jacek Tabarkiewicz
Response to Reviewer 2 Comments
Point 1. Provide additional rationale and evidences in both introduction and discussion why the A/R index is more informative than the levels of adiponectin and resistin alone.
Response 1: Due to the fact that the AR index is little studied, there are few publications related to it. However, as suggested, we have described in more detail what its significance is in the case of obesity and its consequences. However, our study brings new results regarding the impact of lifestyle on this parameter. I have added one more publications that I have not cited. Unfortunately, we have not found more, it is possible that their number is limited.
Point 2. There are typos and other mistakes within the text, which should be corrected after careful proofreading of the entire manuscript. Some of them are:
- You have only one figure, labeled Figure 2, please make a correction or provide rationale for this.
- Line 60 – one comma missing after [7].
- Line 69 – what is that X in brackets for references?
- Row 116 and 119 - p=0042 and p=0044.
- Please use consistently decimal separator - full stop “.”, not comma “,” doublecheck all tables and the text.
- Please, introduce all abbreviations when used for first time.
- On line 120, Supplementary table should be 1, not 2.
- Format the superscripts within the text properly (e.g. m2).
- In subsection 4.4 Height and Body Composition Measurements and Lifestyle and Nutrition Assessment, you should refer to Supplementary Materials: File S1 “Lifestyle and Nutrition Assessment”
Response 2: Thank you for your valuable consideration. We have changed it as per the Reviewer's suggestion.